# Self-Calibration Method and Pose Domain Determination of a Light-Pen in a 3D Vision Coordinate Measurement System

**DOI:** 10.3390/s22031029

**Published:** 2022-01-28

**Authors:** Dongri Shan, Chenglong Zhang, Peng Zhang, Xiaofang Wang, Dongmei He, Yalu Xu, Maohui Zhou, Guoqi Yu

**Affiliations:** 1School of Mechanical Engineering, Qilu University of Technology (Shandong Academy of Sciences), Jinan 250300, China; zcl17864188832@163.com (C.Z.); tyut_qlut_xuyalu@126.com (Y.X.); 1043119049@stu.qlu.edu.cn (M.Z.); 1043119037@stu.qlu.edu.cn (G.Y.); 2School of Electrical Engineering and Automation, Qilu University of Technology (Shandong Academy of Sciences), Jinan 250300, China; zp@qlu.edu.cn (P.Z.); wxfmail2008@163.com (X.W.); he.ferry@163.com (D.H.)

**Keywords:** stylus tip center self-calibration, spherical fitting, pose domain, vision measurement system

## Abstract

Light pens for 3D vision coordinate measurement systems are increasingly widely used due to their advantages, such as their small size, convenience of being carried, and widespread applicability. The posture of the light pen is an important factor that affects accuracy. The pose domain of the pen needs to be given so that the measurement system has a suitable measurement range to obtain more qualified parameters. The advantage of the self-calibration method is that the entire self-calibration process can be completed at the measurement site with no auxiliary equipment. After the system camera calibration was completed, we took several pictures of the same measurement point with different poses to obtain the conversion matrix of the picture and subsequently used spherical fitting, the generalized inverse method of least squares, and the principle of position invariance in the pose domain range. The combined stylus tip center self-calibration method calculates the actual position of the light pen probe. The experimental results verify the effectiveness of the method; the measurement accuracy of the system can satisfy basic industrial measurement requirements.

## 1. Introduction

With the ongoing advancement of human society, manufacturing continues to flourish, and industrial measurement has become an extremely important link in the industrial development process. With the development of the automotive industry and aerospace industry, which require high-precision real-time measurement, there are increasingly more applications and requirements for workpiece measurement, particularly in terms of accuracy. At present, mainstream measurement technology cannot satisfy the ever-increasing measurement requirements. The traditional coordinate measuring machine (CMM) [1] cannot satisfy the online measurement requirements of some industrial sites due to its large size and limited measurement range. Indoor GPS, white light scanners [2,3], optical laser trackers [4,5,6], and other common measurement equipment [7,8] also have many shortcomings such as low measurement efficiency, high cost, complex configuration, and poor portability. The shortcomings of modern measurement technology have limited the development of industry. Therefore, the development of new measurement technology has become a common expectation. Vision measurement systems [9] are mainly based on computer vision, including measurement technology, electronic technology, image processing technology, etc. to create a new measurement structure frame, reduce measurement limitations, increase flexibility and convenience, and enable workpiece measurement from multiple angles. Its small size has met with a favorable response in modern manufacturing fields such as aerospace and automobiles. The light-pen-type 3D vision measurement system [10] based on machine vision is increasingly used because of its small size, convenient portability, flexible assembly according to on-site measurement requirements, and good applicability; and related products have been put into production [11]. Of course, many scholars have unique insights into visual measurement systems and have achieved notable results [12,13,14,15,16,17].

The actual coordinates of the center of the light pen stylus, in the coordinate system, directly affect the measurement accuracy of the three-coordinate measuring system. In practical applications, the center of the light pen stylus is usually taken as a theoretical value when the light pen is used for processing. Due to the processing error of the light pen, the center value directly affects the measurement accuracy. When measuring various measurement objects such as deep holes and grooves, it is necessary to replace the light pen probe with other angles or add an extension rod for measurement. The difference in force and loss of multiple uses of the light pen also cause small changes in the coordinates, so the self-calibration of the stylus tip center is particularly important. Liu optimized an objective function based on the principle of position invariance and generalized an inverse method of the least squares solution of the nonlinear equations and obtained the position of the stylus tip center in the coordinate system [18,19]. Zheng proposed a two-step calibration method for the center of a plane target probe [20]. According to the principle of position invariance, the objective optimization function is established, and the optimized correction value of the stylus tip center in the X and Y directions in the target coordinate system is solved by the Levenberg–Marquardt algorithm [21]. Then, the error of the stylus tip center in the Z direction is solved. Zhang also adopted the principle of position invariance, upon which he provided the error constraint parameters [22].

The posture of the light pen is an important factor that affects the accuracy. The pose domain of the light pen needs to be given so that the measurement system has a suitable measurement range to obtain more qualified parameters. Therefore, a threshold measurement device was designed according to the measurement system in the experiment. The main body was composed of a light pen, connecting rod, base, motor seat, coupling, and other parts. The transmission device was composed of a Siemens servo motor and the corresponding drive. The experiment was completed on the built measurement system [23]. After the motor parameters and program settings were completed, the experiment was driven by the motor to drive the device of the mechanical structure to make precise angle changes. For each rotation of 1°, the corresponding experimental image was collected to obtain the measurement error at this angle, and the data were sorted to obtain the pose that satisfied the requirement area.

A new self-calibration method for the center of the optical pen probe was proposed. The actual position of the light pen probe was obtained by combining methods such as spherical fitting, the generalized inverse method of the least squares method, and the principle of position invariance [19]. Regardless of the changes in pose of the stylus, the position of the center of the stylus remained the same, as shown in Figure 1. After the camera calibration was completed [24,25,26,27], the light pen probe collected multiple sets of images in different poses that satisfied the threshold measurement requirements under the CCD camera and used the conversion relationship of each image to fit the same feature point on a sphere with the center of the probe as the center of the sphere, as shown in Figure 2. Thus, the actual position of the center of the probe was obtained.

## 2. Self-Calibration Method of the Stylus Tip Center

### 2.1. Light-Pen-Type 3D Vision Measurement System

The light-pen-type vision measurement system is mainly composed of three parts: a CCD industrial camera capable of imaging infrared rays, a computer, and a light pen, as shown in Figure 3. The light pen consists of eight infrared LEDs and a detachable ruby ball probe. Figure 4 shows several types of probes. A and B have different core diameters, and the B–E measuring rods have different lengths. During the measurement, the light pen image is taken by the CCD industrial camera and transmitted to the computer for related calculations; finally, the measurement result is obtained.

The measurement process of the measurement system is as follows: After the camera’s internal parameters have been calibrated [28], the user places the measured object and light pen in the range that the CCD camera can shoot, the user presses the shooting button, the CCD camera takes a set of pictures, and the world coordinates of the measured object are obtained. For the coordinates in the system, the distance between two points can be similarly measured by placing the light pen on another point.

### 2.2. Establishment of the Coordinate System

The visual coordinate measurement system contains three coordinate systems:1.Image coordinate system (O_0_-XY):

The coordinate system is established on the imaging surface of the CCD camera with the center of the imaging surface as the coordinate origin. The *X*-axis and *Y*-axis are parallel to the horizontal and vertical imaging directions of the CCD camera, respectively, and parallel to the O_1_UV plane (unit: pixel).

2.Camera coordinate system (O_1_-uvw):

The camera coordinate system is established on the industrial camera, and the origin of the camera coordinate system is the optical center of the industrial camera. The plane formed by the camera coordinate system is also parallel to the O_1_UV plane (unit: mm).

3.Light pen coordinate system (O_2_-xyz):

The theoretical value of the center of the light pen stylus is the coordinate origin, and the direction of the light pen shaft is the *Y*-axis; the lower plane perpendicular to the six characteristic points of the light pen is the *Z*-axis, and the *X*-axis is determined by the right-hand rule (unit: mm).

### 2.3. Solve Equations

Before measurement, the stylus tip center in the light pen coordinate system should be calibrated. During the measurement, the images of the LED targets are captured by a CCD camera, and the computer processes the images to obtain the center coordinates of the center position of the target as a known quantity to calculate the rotation matrix R and translation vector T [23]. The coordinates of the stylus tip center that correspond to the measured position in the camera coordinate system can be obtained by (1). The camera coordinate system is transformed into the light pen coordinate system as follows:$$\left[\begin{array}{c} u\\ v\\ w \end{array}\right]=\left[\begin{array}{cccc} r_{1} & r_{2} & r_{3} & t_{x}\\ r_{4} & r_{5} & r_{6} & t_{y}\\ r_{7} & r_{8} & r_{9} & t_{z} \end{array}\right]\cdot \left[\begin{array}{c} x\\ y\\ z\\ 1 \end{array}\right]$$
(1)$$\left[\begin{array}{ccc} r_{1} & r_{2} & r_{3}\\ r_{4} & r_{5} & r_{6}\\ r_{7} & r_{8} & r_{9} \end{array}\right]=R\left[\begin{array}{c} t_{x}\\ t_{y}\\ t_{Z} \end{array}\right]=T$$

We establish the relationship between the camera coordinate system and pixel coordinate system:(2)$$s\cdot \left[\begin{array}{c} X\\ Y\\ 1 \end{array}\right]=\left[\begin{array}{ccc} f & 0 & 0\\ 0 & f & 0\\ 0 & 0 & f \end{array}\right]\left[\begin{array}{c} u\\ v\\ w \end{array}\right]$$

The conversion between the camera coordinate system and pixel coordinate system is carried out using Equation (2); f is the focal length, $X=\frac{f}{w}u,Y=\frac{f}{w}v$;

We establish the linear equation set of the *j*-th feature point in the image:(3)$$s_{j}\cdot \left[\begin{array}{c} X_{j}\\ Y_{j}\\ 1 \end{array}\right]=t_{z}\left[\begin{array}{cccc} a_{1} & a_{2} & a_{3}^{} & a_{4}\\ a_{5} & a_{6} & a_{7} & a_{8}\\ a_{9} & a_{10} & a_{11} & 1 \end{array}\right]\left[\begin{array}{c} x_{j}\\ y_{j}\\ z_{j}\\ 1 \end{array}\right]$$
let $t_{z}\left[\begin{array}{cccc} a_{1} & a_{2} & a_{3}^{} & a_{4}\\ a_{5} & a_{6} & a_{7} & a_{8}\\ a_{9} & a_{10} & a_{11} & 1 \end{array}\right]=\left[\begin{array}{ccc} f & 0 & 0\\ 0 & f & 0\\ 0 & 0 & f \end{array}\right]\left[\begin{array}{c} u\\ v\\ w \end{array}\right]=\left[\begin{array}{ccc} f & 0 & 0\\ 0 & f & 0\\ 0 & 0 & f \end{array}\right]\left[\begin{array}{cccc} r_{1} & r_{2} & r_{3} & t_{x}\\ r_{4} & r_{5} & r_{6} & t_{y}\\ r_{7} & r_{8} & r_{9} & t_{z} \end{array}\right]$
(4)$$s_{j}=r_{7}x_{j}+r_{8}y_{j}+r_{9}z_{j}+t_{z}=t_{z}(a_{9}x_{j}+a_{10}y_{j}+a_{11}z_{j}+1)=t_{z}\zeta_{j}$$

Equation (3) is equivalent to:(5)$$\zeta_{j}\cdot \left[\begin{array}{c} X_{cj}\\ Y_{cj}\\ 1 \end{array}\right]=\left[\begin{array}{cccc} a_{1} & a_{2} & a_{3}^{} & a_{4}\\ a_{5} & a_{6} & a_{7} & a_{8}\\ a_{9} & a_{10} & a_{11} & 1 \end{array}\right]\cdot \left[\begin{array}{c} x_{j}\\ y_{j}\\ z_{j}\\ 1 \end{array}\right]$$

After solving the linear equations, the parameters are separated to obtain the initial values of R and T.

In the ideal perspective imaging process, the solution obtained by the above formula is feasible because there is no error in the coordinate values of the object point and image point in each coordinate system, and the calculated R satisfies the orthogonal system constraint. However, in the actual environment, due to various factors such as camera parameter calibration error [29], image plane position extraction error of the control point center, and coordinate value calibration error of the control point center in the coordinate system, the R matrix does not satisfy the orthogonal constraint relationship. Consequently, matrix T also has a large error, so the initial values of R and T can be obtained by the linear equation solving method. Then, the optimal solution of R and T can be obtained by the Newton–Gaussian iteration method [23].

### 2.4. Self-Calibration Method of the Stylus Tip Center

After R and T have been determined, because the distance between the characteristic point of the light pen and the actual position of the stylus tip center coordinate is a fixed value, according to the generalized inverse method of the nonlinear least square method, we obtain:(6)$${\left(x_{ij}^{}-x_{0}\right)}^{2}+{\left(y_{ij}^{}-y_{0}\right)}^{2}+{\left(z_{ij}^{}-z_{0}\right)}^{2}=d_{ij}^{2}$$
where *i* represents the *i*-th image, and *j* represents the *j*-th feature point.

The coordinate positions of the pictures in the coordinate system are different, but the distance from the actual position of the center coordinate of the probe is the same. Hence, the same feature point of n groups of pictures in the same spatial coordinate system (based on the probe and actual position of the center coordinate) is the center of the sphere, and the constant distance between the characteristic point and the stylus tip center is the radius of the sphere. The same feature point of all pictures is fitted to a sphere, and eight feature points are fitted to eight spheres. The spherical center coordinates of the fitted spheres are the actual position of the center coordinates of the stylus.

We obtain the residual formula:(7)$$F=\sum_{i=1}^{n}{\left[{\left(x_{i}-x_{0}\right)}^{2}+{\left(y_{i}-y_{0}\right)}^{2}+{\left(z_{i}-z_{0}\right)}^{2}-d^{2}\right]}^{2}$$

The corresponding partial derivatives are
(8)$$\begin{array}{l} \frac{\partial F}{\partial x}=-4\sum_{i=1}^{n}\left(x_{i}-x_{0})[{\left(x_{i}-x_{0}\right)}^{2}+{\left(y_{i}-y_{0}\right)}^{2}+{\left(z_{i}-z_{0}\right)}^{2}-d^{2}\right]\\ \frac{\partial F}{\partial y}=-4\sum_{i=1}^{n}\left(y_{i}-y_{0})[{\left(x_{i}-x_{0}\right)}^{2}+{\left(y_{i}-y_{0}\right)}^{2}+{\left(z_{i}-z_{0}\right)}^{2}-d^{2}\right]\\ \frac{\partial F}{\partial z}=-4\sum_{i=1}^{n}\left(z_{i}-z_{0})[{\left(x_{i}-x_{0}\right)}^{2}+{\left(y_{i}-y_{0}\right)}^{2}+{\left(z_{i}-z_{0}\right)}^{2}-d^{2}\right]\\ \frac{\partial F}{\partial d}=-4d\sum_{i=1}^{n}\left[{\left(x_{i}-x_{0}\right)}^{2}+{\left(y_{i}-y_{0}\right)}^{2}+{\left(z_{i}-z_{0}\right)}^{2}-d^{2}\right] \end{array}$$

We insert the coordinate values, set the partial derivative equal to zero, construct a system of equations, and solve it to obtain the actual position of the center coordinate of the stylus.

### 2.5. Self-Calibration Steps

The self-calibration method is realized on the built visual coordinate measurement system. Briefly, we outline the steps to obtain the actual position of the stylus center coordinates using the self-calibration method.

After the measurement system has been built, use the spatial coordinates of the characteristic points of the light pen measured by CMM as the initial value.Use the principle of position invariance to shoot images of different poses. Considering the particularity of the self-calibration method, the image pose should be changed as much as possible in the threshold range to make the result more accurate.After obtaining at least eight sets of pictures within the threshold range, calculate R and T for each image using Equations (1)–(5).After R and T have been determined, construct and solve Equations (6)–(8).Obtain the actual position of the stylus tip center coordinate in the light pen coordinate system.

## 3. Experiment

### 3.1. Pose Domain Experiment

The rotation angle of the light pen in each direction affects the measurement accuracy. Considering the rotation angles of the three rotation axes of the light pen, the threshold measurement experiment was carried out according to the above method.

We performed the threshold measurement experiment according to the above method.

When the pen was rotated around the *X*-axis, the angle between the light pen and the positive half axis of the *Y*-axis (pitch) was 41°~159°, and there was a clear and recognizable image.

When the pen was rotated around the *Y*-axis, the angle between the light pen and the positive half axis of the *X*-axis (yaw) was 0°~180°, and there was a clear and recognizable image.

When the pen was rotated around the *Z*-axis, the angle between the light pen and the positive half axis of the *X*-axis (roll) was 0°~45° and 136°~180°, and there was a clear and recognizable image.

Due to the processing error of the mechanical device [30,31,32] and measurement error of the measuring equipment, we used 1° as the unit of the data to ensure that the error did not affect the experimental results. The motor drove the light pen to the measurement position, took a stable image, calculated the measurement accuracy, and rotated 1 degree to take the image. Every time it rotated by 1 degree, the camera collected 15 groups of images within the angle, analyzed and processed the data, and compared the obtained measurement results with the real values to determine differences. The graphs are shown in Figure 5, Figure 6 and Figure 7.

When the pitch angle range is 80°~97°, the yaw angle is 80°~101°, and the tilt angle is 0°~2° and 177°~180°, a certain measurement accuracy can be guaranteed to satisfy the measurement requirements. Satisfactory experimental results were obtained.

### 3.2. Self-Calibration Experimental Results

#### 3.2.1. Self-Calibration Measurement Experiment

We carried out the self-calibration experiment according to the method steps in this article. The self-calibration method was incorporated into the light pen measurement system for measurement comparison experiments, and each light pen performed ten measurement experiments. The measured value was compared with the real value (30.0317 mm) to obtain the absolute error and relative error. The measurement results are shown in Table 1.

The measured values of the five light pen styluses are displayed in lines 2–11. The last three lines show the average, absolute error, and relative error. The measurement results of each stylus in the ten images are shown in lines 1–10. As shown in Table 1, the absolute error of the probe of each light pen is stabilized below 0.0737 mm, and the relative error is stabilized below 0.0025 mm. The experimental results satisfy the basic industrial measurement requirements, which verifies the effectiveness of this method.

#### 3.2.2. Single-Point Repeatability Experiment

We put the light pen stylus in a standard cone. To maintain the position of the light pen stylus unchanged, we rotated the position of the light pen to perform a single-point repeatability experiment.

The experimental results of five light pen styluses are shown in Table 2. The coordinates of the center point of each light pen stylus in the ten images are shown in columns 1–10. The single-point repeatability is expressed by the standard deviation of ten center point coordinates. As shown in the previous column, the probe repeatability of each light pen does not exceed 0.041, which verifies the effectiveness of the self-calibration method.

## 4. Conclusions

A self-calibration method for the center of a light pen stylus based on spherical fitting is proposed. Multiple sets of light pen images with different poses were taken, and the actual position of the light pen probe was obtained by combining methods such as spherical fitting, the generalized inverse method of least squares, and the position invariance principle. The model and steps of the self-calibration method are given. To verify the effectiveness of the method, five different light-pen probes were used to conduct experiments, and a single-point repeatability experiment was carried out. The test results show that the measurement accuracy after using the method is obviously better than that before the method is used, and it satisfies the basic industrial measurement requirements.

The pose of the light pen is an important factor that affects the measurement accuracy of the light pen. Combined with the measurement system, the measurement experiment of the light pen pose field was carried out. In the experiment, three angles were measured, and the pose domain of the light pen was obtained, which standardized the pose of the light pen and ensured its measurement accuracy. The next step of research will involve actual industrial applications, and it will be better applied to actual measurements.

## Figures and Tables

**Figure 1 sensors-22-01029-f001:**
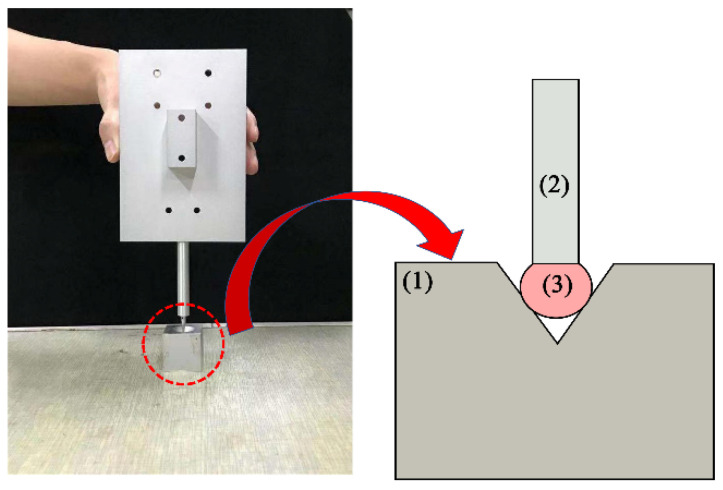
The principle of position invariance: (1) Object to be measured (2) Light pen stylus (3) stylus tip center.

**Figure 2 sensors-22-01029-f002:**
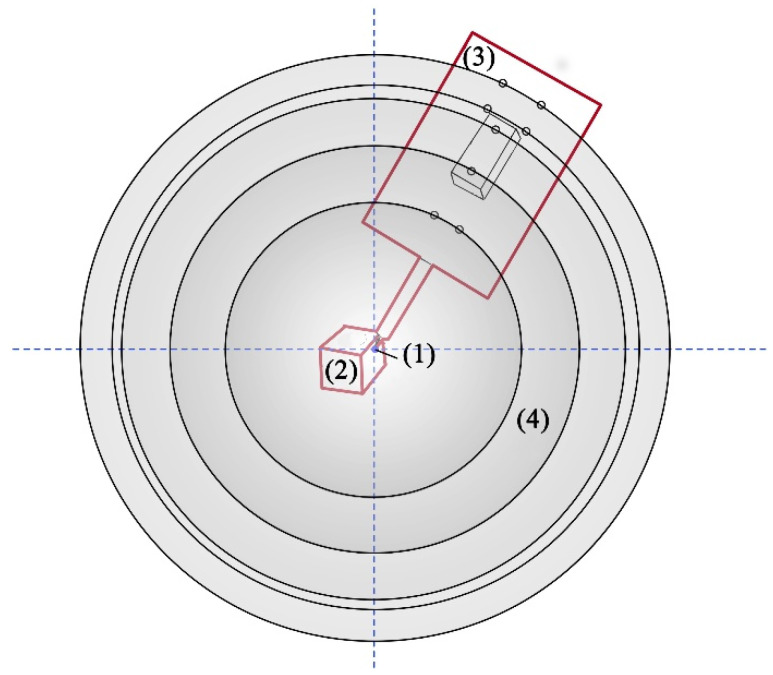
Schematic diagram of the self-calibration method: (1) Stylus tip center (2) The object to be measured (3) The light pen (4) The spherical surface fitted.

**Figure 3 sensors-22-01029-f003:**
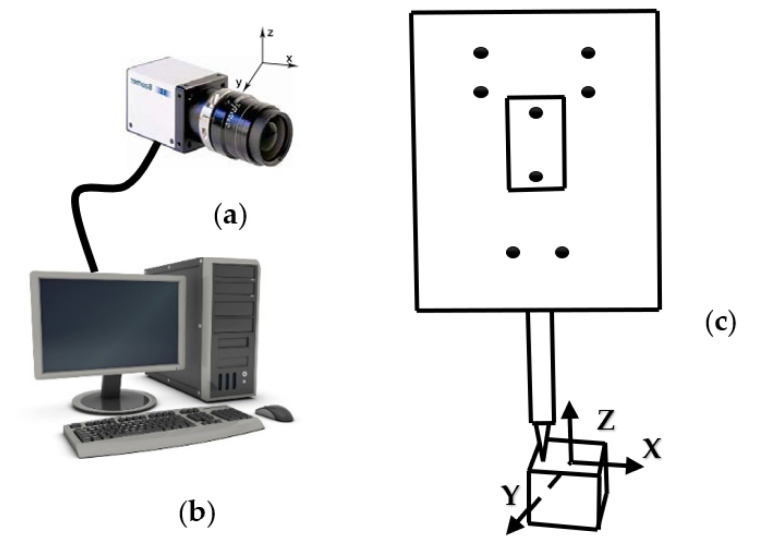
System composition diagram: (**a**) camera; (**b**) computer; (**c**) light pen.

**Figure 4 sensors-22-01029-f004:**
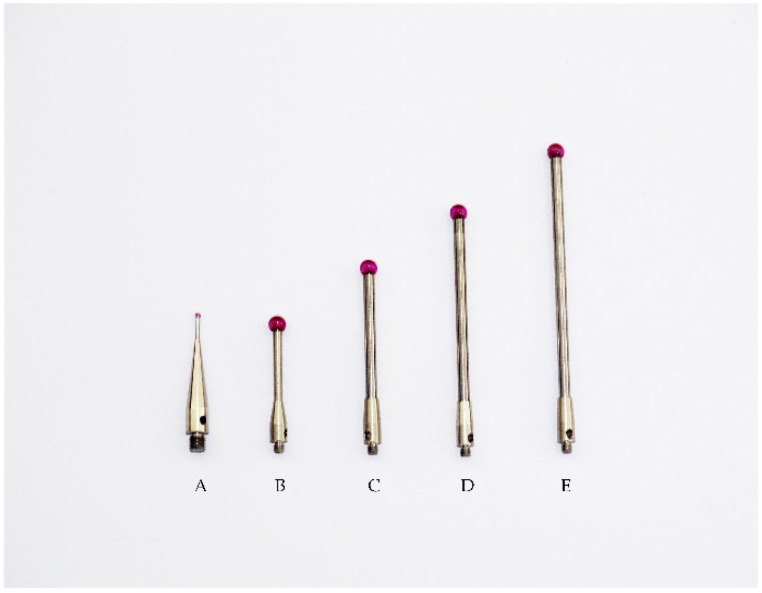
Light pen stylus.

**Figure 5 sensors-22-01029-f005:**
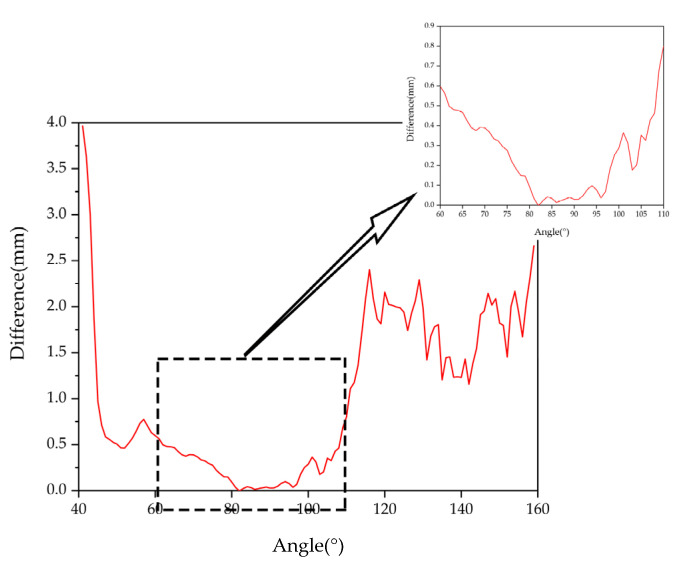
Pitch angle measurement results.

**Figure 6 sensors-22-01029-f006:**
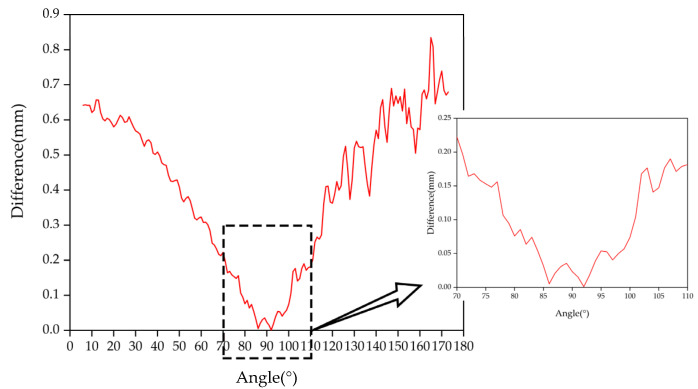
Yaw angle measurement results.

**Figure 7 sensors-22-01029-f007:**
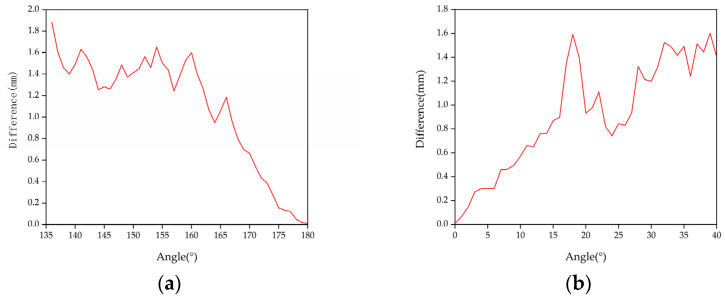
Tilt angle measurement results: (**a**) 0°~40° and (**b**) 136°~180°.

**Table 1 sensors-22-01029-t001:** Self-calibration experiment.

Stylus Type	A	B	C	D	E
1	29.989	30.331	30.272	30.242	29.923
2	29.870	30.305	30.248	30.214	29.853
3	30.083	30.294	30.254	30.244	30.058
4	29.733	29.865	29.828	29.820	29.750
5	29.905	29.792	29.770	29.794	29.939
6	30.135	30.058	30.051	30.080	30.127
7	29.994	29.773	29.764	29.801	30.018
8	29.877	29.560	29.561	29.609	29.869
9	30.187	29.892	29.900	29.956	30.209
10	30.036	29.948	29.931	29.958	30.046
Average value	29.981	29.982	29.958	29.972	29.979
Absolute error	0.051	0.050	0.074	0.060	0.053
Relative error	0.17%	0.17%	0.25%	0.20%	0.18%

**Table 2 sensors-22-01029-t002:** Single-point repeatability experiment.

	Test	1	2	3	4	5	6	7	8	9	10	AVE	STD
A	u	78.995	78.995	79.014	79.004	78.997	78.997	78.983	79.002	78.990	78.995	78.997	0.008
	v	79.967	79.975	79.965	79.968	79.961	79.964	79.982	79.956	79.979	79.980	79.970	0.009
	w	790.029	789.994	790.040	790.026	790.047	790.030	789.965	790.083	789.986	789.948	790.015	0.041
B	u	79.061	79.074	79.054	79.066	79.068	79.071	79.056	79.064	79.068	79.047	79.063	0.008
	v	80.262	80.250	80.275	80.247	80.261	80.243	80.268	80.260	80.271	80.269	80.261	0.011
	w	790.209	790.262	790.205	790.261	790.220	790.271	790.236	790.277	790.216	790.219	790.238	0.028
C	u	79.094	79.096	79.090	79.094	79.103	79.094	79.093	79.102	79.096	79.097	79.096	0.004
	v	80.626	80.604	80.619	80.634	80.616	80.625	80.627	80.611	80.628	80.609	80.620	0.01
	w	790.286	790.298	790.263	790.222	790.283	790.235	790.231	790.302	790.250	790.289	790.266	0.03
D	u	78.907	78.900	78.897	78.900	78.906	78.909	78.899	78.887	78.899	78.906	78.901	0.006
	v	79.347	79.352	79.352	79.348	79.340	79.340	79.345	79.357	79.349	79.342	79.347	0.005
	w	789.472	789.444	789.425	789.474	789.508	789.518	789.481	789.404	789.467	789.509	789.470	0.037
E	u	79.155	79.166	79.166	79.159	79.178	79.167	79.175	79.167	79.164	79.169	79.167	0.007
	v	80.843	80.869	80.860	80.845	80.861	80.859	80.856	80.860	80.868	80.883	80.860	0.012
	w	790.859	790.855	790.882	790.908	790.881	790.845	790.909	790.907	790.864	790.843	790.875	0.026

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
