# Peer review of "Self-Calibration Method and Pose Domain Determination of a Light-Pen in a 3D Vision Coordinate Measurement System"

_sensors, 2022, doi:10.3390/s22031029_

Round 1

Reviewer 1 Report

The paper deals with self-calibration of a light pen, which is emerging research topic fully in the scope of the journal. The structure of the paper is clear and the language quality appropriate for the journal. However, there are few drawbacks to be mentioned:

1) Quality and resolution of all Figures should be checked (see e.g. Fig 5 and 6)
2) I guess the light pen stylus on Fig. 4 can be denoted as A.. E as the referred in Table 1 
3) It seems the math background is more or less adopted from previous works. If there are significant novelties, they are not sufficiently highlighted in the text.
4) It would be nice to compare the results obtained (KPIs) also with other methods referred in Introduction. 
5) I think Fig. 1 and 2 can be improved, perhaps some notation can be added. Just in order to make them more self-understandable.
6) It seems Fig 1 is not referred in the text, I propose to doublecheck those references and order in which they appear in the text.
7) I think it is not necessary to refer to errors achieved (e.g. 0.0737) in the abstract. Abstract often appears as independent text in databases and those numbers are difficult to understand without full paper context.
8) I would unify the style of equations, e.g. italic vs normal font, etc.
9) Conclusion could outline also some future plans and pathways towards real applications
10) Perhaps final grammar and spell check should be done, some sentences can be polished e.g. "When measuring various measurement objects ---"
11) I would also doublecheck if all methods are properly referred, e.g. some citations for "Levenberg–Marquardt algorithm" should be added.
12) The Introduction can be finished by brief overview of the content of the rest of the papers with references to sections, sub-sections.
13) The sub-section title "Parameter solving" sounds a bit strange to me. One can solve equations etc.
14) Inserting matrices directly into the text makes it less readable (check line 144). I would change this somehow.
15) Also the title of 3.2 is a bit confusing, it seems not the method but experimental results are presented in this part.

Most of the comments are just formal. In principle the paper can be published after a revision, final formatting and provided that all reviewers will agree.

Author Response

Response to Reviewer

Point 1: Quality and resolution of all Figures should be checked (see e.g. Fig 5 and 6)

Response 1: Thank you very much for your comment. All Figures resolutions have been resized to high resolution (dpi at least 300)

Point 2: I guess the light pen stylus on Fig. 4 can be denoted as A.. E as the referred in Table 1

Response 2: Thank you very much for your comment. The light pen stylus on Fig. 4 has been denoted A.. E, and their differences are expressed.

Point 3: It seems the math background is more or less adopted from previous works. If there are significant novelties, they are not sufficiently highlighted in the text.

Response 3: Thank you very much for your comment. The innovation in the math background has been reflected in another article of mine (DOI: 10.1109/ACCESS.2021.3079274), the main innovation of this paper is a new self-calibration method. In the follow-up research I will focus on the innovation of mathematics.

Point 4: It would be nice to compare the results obtained (KPIs) also with other methods referred in Introduction.

Response 4: Thank you very much for your comment. Differences between methods cannot be compared because the measurement systems used are different. In the following research, I will give a comparison of different methods based on the same measurement system.

Point 5: I think Fig. 1 and 2 can be improved, perhaps some notation can be added. Just in order to make them more self-understandable.

Response 5: Thank you very much for your comment. Fig. 1 and 2 have been improved for ease of understanding. Detailed revisions can be seen in the revised manuscript.

Point 6: It seems Fig 1 is not referred in the text, I propose to doublecheck those references and order in which they appear in the text.

Response 6: Thank you very much for your comment. References have been carefully checked and the order in which they appear in the text, and corrections have been made. Detailed revisions can be seen in the revised manuscript.

Point 7: I think it is not necessary to refer to errors achieved (e.g. 0.0737) in the abstract. Abstract often appears as independent text in databases and those numbers are difficult to understand without full paper context.

Response 7: Thank you very much for your comment. The abstract has been changed and the error data has been removed.

Point 8: I would unify the style of equations, e.g. italic vs normal font, etc.

Response 8: Thank you very much for your comment. The style of the equations has been unified, using Palatino Linotype font, size 10, and italic.

Point 9: Conclusion could outline also some future plans and pathways towards real applications.

Response 9: Thank you very much for your comment. The conclusions have been revised and detailed revisions can be seen in the revised manuscript.

Point 10: Perhaps final grammar and spell check should be done, some sentences can be polished e.g. "When measuring various measurement objects ---".

Response 10: Thank you very much for your comment. To ensure grammatical correctness, we have edited the English language, grammar, punctuation, spelling and overall style using professional editors.

Point 11: I would also doublecheck if all methods are properly referred, e.g. some citations for "Levenberg–Marquardt algorithm" should be added.

Response 11: Thank you very much for your comment. All method citations have been checked and detailed revisions can be seen in the revised manuscript.

Point 12: The Introduction can be finished by brief overview of the content of the rest of the papers with references to sections, sub-sections.

Response 12: Thank you very much for your comment. The Introduction section has been revised and detailed revisions can be seen in the revised manuscript.

Point 13: The sub-section title "Parameter solving" sounds a bit strange to me. One can solve equations etc.

Response 13: Thank you very much for your comment. The sub-section title has been changed to “Solve equations”.

Point 14: Inserting matrices directly into the text makes it less readable (check line 144). I would change this somehow.

Response 14: Thank you very much for your comment. The matrix and text have been separated and detailed revisions can be seen in the revised manuscript.

Point 15: Also the title of 3.2 is a bit confusing, it seems not the method but experimental results are presented in this part.

Response 15: Thank you very much for your comment. 3.2 The title has been changed to”Self-calibration experimental results” .

Reviewer 2 Report

Dear authors, please find below some suggestions that must be considered for improvement of the manuscript ‘Self-calibration Method and Pose Domain Determination of Light-Pen in a 3D Vision Coordinate Measurement System’, Manuscript ID: sensors-1564259, as follows:

  1. Considering measurement of objects (details) containing deep/wide whole/dimples (lines 61-63), there were many studies provided previously for reducing the measurement and data analysis errors. There are only a few wors around measurement error and processing errors of the mechanical device (lines 213-215). Please try to emphasize your paper against currently published studies and, simultaneously, make a review of the papers with a more critical, e.g.:

(1) https://doi.org/10.1016/j.wear.2007.01.108

(2) https://doi.org/10.3390/ma14154077

(3) https://doi.org/10.1016/j.cirpj.2021.03.016

  1. Authors should mention some recent (from the last 3-4 years) papers, published in the Sensor journal, considering the analysis of CMM techniques, e.g.

(4) https://doi.org/10.3390/s19245346

(5) https://doi.org/10.3390/s19122667

  1. Sentences from lines 76-95 should be moved to other section. Some, the introduction of the method applied, respectively, should be located in the “introduction’ section, nevertheless, only a few sentences should be placed in this part. I suggest moving this sentence for section 2 and, if required, create another, first additional section.
  2. Figure 4 was not mentioned in the text. All of the figures and tables must be cited, otherwise are redundant.
  3. The novelty should be highlighted that most of the proposals were introduced as presented previously, already published by the authors. It looks like the authors propose an analysis based on already used techniques. From that point of view, the manuscript seems to be more technical than scientific. Where is the novelty? Please, try to emphasize what is new in the studies presented.
  4. What is the difference, except the length for probes 2-5 (counting from the left to right in Figure 4)? Maybe it should be mentioned that there were two types of probes but, respectively, some more with different lengths, or other sentences.
  5. There is no reference no 33 – line 109. Whereas the sentence from lines 160-169 is based on previous studies it should be mentioned (cited).
  6. How was the number of coordinates (8n of n sets…, lines 182-183) selected? Arbitrary? It should be justified, even commonly known, for a regular reader.
  7. It should be introduced (in section 2.5) the steps, not starting with them straightly. Similarly, in section 3.1. when 3 axes of rotating are considered.
  8. ‘Conclusion’ section was written unsatisfactorily, at least, for a regular reader. Even containing the most crucial data, seems too short or, remarking, too general. Please try to add some more valuable information. Further, try to divide the conclusion into numbered parts that, can be helpful for novelty highlighting as well.

Moreover, some general, editorial issues, were found:

  1. In references there must be unified that some authors are written with the first author only and ‘et.al’ and, respectively, some are presented with all of the authors. Unification must be provided. The current state is messy and unclear.
  2. Citing the Sensors journal, as far as concerned, there are no requirements for adding a (Basel) but can be falsy informed.
  3. All of the authors should be cited with surname firstly and then the first letter of names, it was found not unified, e.g. in lines 289 or 299.
  4. There is no gap (space) between the text and cited items (references in the text), e.g. lines 46, 54, 56.
  5. Feels like the sentences from lines 251-255, should be re-located before Table 2. Usually, comments should be placed before some results presented in figures or, respectively, tables. The results should not outstrip (exceed) their comments, it is difficult for a reader and, unfortunately, confuse him.

Generally, the study area is interesting, however, the manuscript, at least in its current form, is confusing and, in some cases, messy, therefore not suitable for taking into consideration to be published in the Sensors journal.

Concluding, improvements, at least those suggested above, must be provided.

Author Response

Response to Reviewer

Point 1: Considering measurement of objects (details) containing deep/wide whole/dimples (lines 61-63), there were many studies provided previously for reducing the measurement and data analysis errors. There are only a few wors around measurement error and processing errors of the mechanical device (lines 213-215). Please try to emphasize your paper against currently published studies and, simultaneously, make a review of the papers with a more critical, e.g.:

(1) https://doi.org/10.1016/j.wear.2007.01.108

(2) https://doi.org/10.3390/ma14154077

(3) https://doi.org/10.1016/j.cirpj.2021.03.016

Response 1: Thank you very much for your comment. The manuscript has been briefly revised in response to your comments. In the following research, we will try to make a detailed analysis on the measurement error and processing error of mechanical equipment to improve the measurement accuracy of the measurement system. Thanks again for your suggestion.

Point 2: Authors should mention some recent (from the last 3-4 years) papers, published in the Sensor journal, considering the analysis of CMM techniques, e.g.

(4) https://doi.org/10.3390/s19245346

(5) https://doi.org/10.3390/s19122667

Response 2: Thank you very much for your comment. The papers given have already been mentioned in the manuscript, respectively refs [16][17].

Point 3: Sentences from lines 76-95 should be moved to other section. Some, the introduction of the method applied, respectively, should be located in the “introduction’ section, nevertheless, only a few sentences should be placed in this part. I suggest moving this sentence for section 2 and, if required, create another, first additional section.

Response 3: Thank you very much for your comment. Sentences have been adjusted to other parts. Detailed revisions can be seen in the revised manuscript.

Point 4: Figure 4 was not mentioned in the text. All of the figures and tables must be cited, otherwise are redundant.

Response 4: Thank you very much for your comment. All figures and tables have been cited and detailed revisions can be seen in the revised manuscript.

Point 5: The novelty should be highlighted that most of the proposals were introduced as presented previously, already published by the authors. It looks like the authors propose an analysis based on already used techniques. From that point of view, the manuscript seems to be more technical than scientific. Where is the novelty? Please, try to emphasize what is new in the studies presented.

Response 5: Thank you very much for your comment. The main innovation of this paper is mainly a new self-calibration method, and the detailed modification can be seen in the revised manuscript. In the follow-up research I will focus on the innovation of mathematics.

Point 6: What is the difference, except the length for probes 2-5 (counting from the left to right in Figure 4)? Maybe it should be mentioned that there were two types of probes but, respectively, some more with different lengths, or other sentences.

Response 6: Thank you very much for your comment. The light pen stylus has been denoted A..E, and their differences are expressed. Detailed revisions can be seen in the revised manuscript.

Point 7: There is no reference no 33 – line 109. Whereas the sentence from lines 160-169 is based on previous studies it should be mentioned (cited).

Response 7: Thank you very much for your comment. References have been carefully checked and the order in which they appear in the text, and corrections have been made. Detailed revisions can be seen in the revised manuscript.

Point 8: How was the number of coordinates (8n of n sets…, lines 182-183) selected? Arbitrary? It should be justified, even commonly known, for a regular reader.

Response 8: Thank you very much for your comment. The choice of coordinates has been rationally explained and detailed revisions can be seen in the revised manuscript.

Point 9: It should be introduced (in section 2.5) the steps, not starting with them straightly. Similarly, in section 3.1. when 3 axes of rotating are considered.

Response 9: Thank you very much for your comment. An introduction to the steps has been added to the manuscript and detailed revisions can be seen in the revised manuscript.

Point 10: “Conclusion”section was written unsatisfactorily, at least, for a regular reader. Even containing the most crucial data, seems too short or, remarking, too general. Please try to add some more valuable information. Further, try to divide the conclusion into numbered parts that, can be helpful for novelty highlighting as well.

Response 10: Thank you very much for your comment. The "Conclusion" section has been revised. An attempt has been made to divide into two parts and add some information to highlight the novelty. Detailed revisions can be seen in the revised manuscript.

Point 11: In references there must be unified that some authors are written with the first author only and ‘et.al’ and, respectively, some are presented with all of the authors. Unification must be provided. The current state is messy and unclear.

Response 11: Thank you very much for your comment. The author format in references has been unified to write only the first author and "et al."

Point 12: Citing the Sensors journal, as far as concerned, there are no requirements for adding a (Basel) but can be falsy informed.

Response 12: Thank you very much for your comment. Citations to Sensors journals have been added (Basel).

Point 13: All of the authors should be cited with surname firstly and then the first letter of names, it was found not unified, e.g. in lines 289 or 299.

Response 13: Thank you very much for your comment. Citations for all authors have been unified (surname firstly and then the first letter of names).

Point 14: There is no gap (space) between the text and cited items (references in the text), e.g. lines 46, 54, 56.

Response 14: Thank you very much for your comment. Both the main text and the cited items (references in the main text) have been checked and gaps added.

Point 15: Feels like the sentences from lines 251-255, should be re-located before Table 2. Usually, comments should be placed before some results presented in figures or, respectively, tables. The results should not outstrip (exceed) their comments, it is difficult for a reader and, unfortunately, confuse him.

Response 15: Thank you very much for your comment. All annotations have been placed before the tables and detailed revisions can be seen in the revised manuscript.

Round 2

Reviewer 1 Report

The second version of the paper has significantly higher quality. Authors responded properly to all comments. Here is the check:

1) Quality and resolution of all Figures 
   OK, improved.
2) OK
3) Math background: primarily adopted from other work. Although explained, it decreases a bit a scientific novelty.
4) Comparison of other methods: 
   Is planned as future research. This decreases a bit a scientific soundness of this paper.
5) OK
6) OK
7) OK
8) OK
9) OK
10) OK
11) OK
12) OK
13) OK
14) OK
15) OK

Despite minor drawbacks, I think the paper can be published provided that all reviewers agree (after final formating etc.)

Reviewer 2 Report

Dear Authors, according to the review of the revised manuscript ‘Self-calibration Method and Pose Domain Determination of Light-Pen in a 3D Vision Coordinate Measurement System’, Manuscript ID: sensors-1564259, it was found suitable improved and, therefore, can be considered for publication in the Sensors journal.

Thank you for taking into consideration the suggested remarks. All of the responses were presented in a required manner.

Good luck in further studies.